# Supersaturation-Dependent Formation of Amyloid Fibrils [note 1]

**DOI:** 10.3390/molecules27144588

**Published:** 2022-07-19

**Authors:** Yuji Goto, Masahiro Noji, Kichitaro Nakajima, Keiichi Yamaguchi

**Affiliations:** 1Global Center for Medical Engineering and Informatics, Osaka University, 2-1 Yamadaoka, Suita 565-0871, Japan; 2Graduate School of Human and Environmental Studies, Kyoto University, Kyoto Sakyo-ku, Kyoto 606-8316, Japan

**Keywords:** amyloid fibrils, amorphous aggregation, amyloid β, β2-microglobulin, protein misfolding, solubility, supersaturation, ultrasonication

## Abstract

The supersaturation of a solution refers to a non-equilibrium phase in which the solution is trapped in a soluble state, even though the solute’s concentration is greater than its thermodynamic solubility. Upon breaking supersaturation, crystals form and the concentration of the solute decreases to its thermodynamic solubility. Soon after the discovery of the prion phenomena, it was recognized that prion disease transmission and propagation share some similarities with the process of crystallization. Subsequent studies exploring the structural and functional association between amyloid fibrils and amyloidoses solidified this paradigm. However, recent studies have not necessarily focused on supersaturation, possibly because of marked advancements in structural studies clarifying the atomic structures of amyloid fibrils. On the other hand, there is increasing evidence that supersaturation plays a critical role in the formation of amyloid fibrils and the onset of amyloidosis. Here, we review the recent evidence that supersaturation plays a role in linking unfolding/folding and amyloid fibril formation. We also introduce the HANABI (HANdai Amyloid Burst Inducer) system, which enables high-throughput analysis of amyloid fibril formation by the ultrasonication-triggered breakdown of supersaturation. In addition to structural studies, studies based on solubility and supersaturation are essential both to developing a comprehensive understanding of amyloid fibrils and their roles in amyloidosis, and to developing therapeutic strategies.

## 1. Supersaturation

Supersaturation is a fundamental natural principle, determining the phases (i.e., vapor, liquid, and solid) of substances [1,2,3,4,5,6]. Supersaturation of a solution refers to a non-equilibrium phase in which, although the solute’s concentration is higher than its thermodynamic solubility, the solute molecules remain soluble for an extended period because of a high free-energy barrier to nucleation. Upon nucleation and crystal formation, the concentration of the solute decreases to its thermodynamic solubility. The supersaturation phenomenon is common in the temperature-dependent liquid–solid or vapor–liquid phase transitions of substances, with one well-known example being the super-cooling of water prior to ice formation. In general, supersaturation is required for the formation of crystals [2]. Supersaturation has been shown to play a role in other ordered protein associations, such as in the fiber formation of hemoglobin S, the molecular basis of sickle cell anemia [7,8]. Supersaturation also plays a role in various types of lithiasis, including urolithiasis, cholelithiasis, and gout, where crystals of calcium oxalate, cholesterol, and uric acid are formed, respectively [9].

A general phase diagram of protein solubility dependent on the precipitant concentration illustrates the role of supersaturation (Figure 1) [3,5,10,11,12]. The phase diagram consists of a soluble region (Region I), a metastable region (Region II), a labile region (Region III), and an amorphous region (Region IV). Below solubility (Region I), monomers are thermodynamically stable. In the metastable region (Region II), supersaturation is thought to persist in the absence of seeding or agitation. In the labile region (Region III), spontaneous nucleation occurs after a certain lag time. Finally, the amorphous region (Region IV) is dominated by amorphous aggregation occurring without a lag time, possibly by the concomitant formation of many nuclei. It is important to note that, after crystallization or amorphous aggregation, the soluble protein concentration becomes equal to the solubility.

Supersaturation is quantified in several ways; often used are the supersaturation ratio (S) and the degree of supersaturation (σ) [2,5]:(1)$$S=\left[C\right]/{\left[C\right]}_{C}$$
(2)$$\sigma =\left(\left[C\right]-{\left[C\right]}_{C}\right)/{\left[C\right]}_{C}$$
where [C] and [C]_C_ are the initial solute’s concentration and thermodynamic solubility, respectively. S and σ are 1 and 0, respectively, at the boundary between Regions I and II, and they increase in Region III with an increase in precipitant concentration (i.e., the driving force of precipitation). In Region IV, amorphous aggregation without a lag time makes evaluations of S or σ difficult. The driving force of nucleation for crystallization is assumed to be proportional to lnS [2]. It is also assumed that the nucleation rate is proportional to the inverse of lag time [13,14,15]. 

The physicochemical mechanisms underlying supersaturation have been studied extensively. One of the classical mechanisms underlying supersaturation is the difficulty of nucleation, as modeled for actin polymerization by Oosawa and Kasai [16]; another is classical nucleation theory [17]. However, subsequent studies suggest a more complicated mechanism of supersaturation, in which solutes form a kinetically trapped state, which is located on the distinct pathway(s) to the formation of crystals. Using molecular dynamics simulations, Matsumoto et al. [18,19] showed that the microscopic homochiral domains, which are more energetically stable than average but cannot grow into macroscopic ice structures due to geometrical frustrations, are the major constituent of supercooled liquid water. Studies with small-angle X-ray scattering [20,21], small-angle neutron scattering [22], and dynamic light scattering [23,24] have suggested that equilibrium clusters or networks of solutes with increased stability play a role in maintaining supersaturation. Matsushita et al. [25,26] studied the solution structures of supersaturated sodium acetate and hen egg white lysozyme (HEWL) with diffracted X-ray tracking, and showed that nanoscale networks with increased dynamics and stability are responsible for supersaturation phenomena. These studies generated a new view of supersaturation: it is a kinetically trapped “stable” state, rather than a state before the high barrier of nucleation, as was suggested by the classical actin polymerization model [16]. 

On other hand, recent work on crystal nucleation focuses on highly concentrated disordered droplets observed before crystallization [4,5,27,28]. More recently, Yamazaki et al. [29] used time-resolved liquid-cell transmission electron microscopy (TEM) to perform an in situ observation of the nucleation of HEWL native crystals. Their TEM images revealed that amorphous solid particles act as heterogeneous nucleation sites. Their findings represent a significant departure from the two-step nucleation and growth mechanism, and suggest that amorphous particle-dependent heterogeneous nucleation is the dominant mechanism of spontaneous crystallization under supersaturation. Amorphous aggregate-assisted amyloid formation was suggested under conditions where oligomer formation is a rare event [30]. We also observed the amorphous oligomeric aggregate-dependent acceleration of HEWL amyloid formation under supersaturation [31]. Taken together, these studies demonstrate that, although supersaturation is an evident phenomenon, there is still uncertainty as to the physicochemical mechanisms that underlie the development, retention, and breakdown of supersaturation, as well as the role of supersaturation in crystal nucleation.

## 2. Amyloid Fibril Formation

### 2.1. Similarity with Crystal Formation

As reviewed previously [6], soon after the discovery of prion phenomena such as kuru and scrapie diseases, it was recognized that prion disease transmission and propagation share some similarities with the process of crystallization [32]. In other words, supersaturation was considered to be an important factor underlying prion diseases. Subsequent studies exploring the structural and functional association between amyloid fibrils and amyloidosis solidified this developing paradigm [6,33,34]. The key pieces of evidence supporting this link were that: (i) spontaneous amyloid formation occurs by a nucleation and growth mechanism, requiring a long lag time, (ii) seeding is an efficient approach for accelerating amyloid formation by escaping the high energy barrier associated with nucleation, and (iii) amorphous aggregates predominate under solute conditions well above solubility.

A useful literary analogy that captures the similarities between amyloid fibril formation and crystallization has been [35] drawn from Kurt Vonnegut’s novel *Cat’s Cradle* [36]. In this fictional work, ice 9, a high temperature ice nucleus (stable at 45.8 °C) is created to help troops escape from mud; however, it is revealed to be an Armageddon device: once a small block of ice 9 is exposed to water, it starts a world-wide water ‘freeze’ due to seed-dependent propagation.

However, recent studies have not necessarily focused on supersaturation, possibly because of marked advancements in structural studies clarifying the atomic structures of amyloid fibrils based on the X-ray crystallographic, solid-state NMR, and cryoEM approaches [37,38,39,40,41,42]. Now, the mechanisms of amyloid formation, distinct disease phenotypes, and familial amyloidosis are discussed, based on the atomic structures of amyloid fibrils, their polymorphs, and the effects of mutations on amyloid structures.

On the other hand, there is increasing evidence that supersaturation plays a critical role in the onset of amyloidosis. Investigations into the temporal evolution of major biomarkers of Alzheimer’s disease [43,44] in the onset and progression of clinical symptoms have shown that concentrations of amyloid β(1-42) peptide (Aβ(1-42)) in the cerebrospinal fluid (CSF) decline 25 and 15 years before expected symptom onset and Aβ(1-42) deposition, respectively. Similar decreases in the concentration of α-synuclein (αSN) were reported for Parkinson’s disease [45,46,47,48,49]. Here, the supersaturation hypothesis provides a simple physical explanation as to why CSF Aβ(1-42) or αSN decreases concomitantly with the deposition of amyloid fibrils [6,50]. The increase in the Aβ(1-40) / Aβ(1–42) ratio for plasma of individuals with Alzheimer’s disease [51] might also be explained by the supersaturation hypothesis: the breakdown of supersaturation decreases the soluble concentration of Aβ(1-42) more than that of Aβ(1-40). Moreover, the effects of polyphosphates [52,53], strong acids [54], the isoelectric point [55], or other additives [15,56] indicate that various conditions enhance the driving force of amyloid formation, and that the breakdown of supersaturation is essential to trigger this reaction. Vendruscolo and colleagues addressed the role of supersaturation from a different viewpoint [57,58], stating that the proteins most at risk for aggregation within the cell are those with high expression levels with respect to their solubilities. To achieve a comprehensive understanding of amyloid fibrils and to develop therapeutic strategies, studies based on solubility and supersaturation are essential. This article considers the progress made since the review article published in 2016 [6] on the role of supersaturation in amyloid fibril formation. 

### 2.2. Supersaturation-Veiled Amyloid Formation Revealed by Heating under Agitation

β2-microglobulin (β2m), a globular β-barrel protein with 99 amino acid residues and an immunoglobulin fold, is responsible for dialysis-related amyloidosis [59,60,61]; it is one of the most extensively studied amyloidogenic proteins, because it allows for the detailed study both of protein unfolding/folding and of amyloid fibril formation [39,62,63,64,65,66,67,68,69,70,71]. Although amyloid deposits of β2m in dialysis patients are observed at a neutral pH, amyloid formation in vitro has been difficult to detect at a neutral pH because of its resistant native structure [11,62,72]. To further understand the mechanism of amyloid formation in vivo, Noji et al. [12] investigated the association between protein folding/unfolding and misfolding leading to amyloid formation (Figure 2).

The researchers examined the heat denaturation of β2m with or without stirrer agitation; they also monitored amyloid formation via the amyloid-specific thioflavin T (ThT) fluorescence, and the total amount of aggregates via light scattering (LS). They found that β2m efficiently forms amyloid fibrils even at a neutral pH by heating under agitation. They constructed temperature- and NaCl concentration-dependent conformational phase diagrams in the presence or absence of agitation (Figure 3), illustrating how amyloid formation under neutral pH conditions is related to thermal unfolding and the breakdown of supersaturation. 

Before the breakdown of supersaturation, a “protein concentration-independent” two-state mechanism applies.
D ⇌ N (Mechanism 1),

The equilibrium constant (*K*_N_) and Gibbs free energy change of folding (Δ*G*_N_) are given by Equations (3) and (4), respectively.
(3)$$K_{N}=\left[N\right]/\left[D\right]$$
(4)$$∆G_{N}=-RT\ln K_{N}$$

Here, *R* and *T* are the gas constant and the temperature *T* in Kelvin, respectively. Consideration of the temperature dependences of the thermodynamic parameters (i.e., Δ*G*_N_, enthalpy change (Δ*H*_N_), and entropy change (Δ*S*_N_)) leads to the temperature-dependent changes in fractions of [N] and [D] (Figure 3, see [12] in detail). The combined effects of entropy and enthalpy terms lead to heat- and cold-denaturations [73], although cold-denaturation does not occur for β2m above 0 °C.

Although the detailed mechanisms of amyloid formation remain elusive [14], a simplified model (Mechanism 2) is valid for describing the equilibrium between monomers (D) and polymeric amyloid fibrils (P) [33,34,62,64]:P + D ⇌ P  (Mechanism 2)

The elongation of fibrils is defined by the equilibrium association constant (*K*_Pol_) as:(5)$$K_{\mathrm{Pol}}=\frac{\left[P\right]}{\left[P\right]\left[D\right]}\;$$

The equilibrium is independent of the molar concentration of amyloid fibrils, [P]. Hence, the equilibrium monomer concentration [D]_C_ is:(6)$${\left[D\right]}_{C}=\frac{1}{K_{\mathrm{Pol}}}\;$$

[D]_C_ is referred to as the “critical concentration” [33,34,64,74] because amyloid fibrils form when the concentration of monomers exceeds [D]_C_. It is noted that [D]_C_ corresponds to [C]_C_ in Equations [1] and [2]. By determining [D]_C_, the apparent free energy change of amyloid formation (ΔG_Pol_) is obtained by:(7)$$∆G_{\mathrm{Pol}}=-RT\ln K_{\mathrm{Pol}}=RT\ln{\left[D\right]}_{C}$$

Assuming that the Gibbs free energy equations (i.e., Δ*G*_Pol_(*T*), Δ*H*_Pol_(*T*), and Δ*S*_Pol_(*T*)) also hold true for polymeric amyloid fibrils [12,64,74], and that the heat capacity change upon amyloid formation is the same as that of protein folding, temperature-dependent changes of Δ*G*_N_ and fractions of [N], [D], and [P] can be obtained (Figure 3, see [12] for detail). As is the case for the native state, heat- and cold-denaturations of amyloid fibrils are expected.

Upon the breakdown of supersaturation, a three-state mechanism between the native, unfolded, and “protein concentration-dependent” amyloid states determines the overall equilibrium. The transition from the two-state mechanism to the three-state mechanism shifts the overall equilibrium in the direction of amyloid fibrils, apparently destabilizing the native state by the law of mass action (Figure 3B). The results suggest that heating and agitation play important roles in the onset of amyloidosis.

One of the most important findings of a series of studies by Goto and colleagues is that supersaturation-dependent amyloid formation is comprehensively and exactly understood by combining the unfolding/refolding conformational transition under supersaturation, and amyloid formation after the breakdown of supersaturation (Figure 3). As long as we consider monomeric proteins, the former is independent of protein concentration, while the latter is critically dependent on protein concentration (i.e., solubility). Although many studies have addressed the relationship between protein folding and misfolding and schematic folding funnels or energy landscapes [75,76,77], the exact unification of these two processes, which is in fact simple, has not previously been presented.

### 2.3. Generality of the Supersaturation-Limited Amyloid Formation

To address the generality of the link between reversible unfolding/refolding under supersaturation and amyloid formation after the breakdown of supersaturation as revealed by β2m, Noji et al. [78] examined the heat denaturation of various proteins with or without stirrer agitation. The study included not only typical amyloidogenic proteins, but also several textbook proteins used previously in folding/unfolding studies (Figure 2).

According to their aggregation behavior, three types of proteins can be defined.

Type S proteins: The first type of protein shows a strict dependence on agitation for amyloid formation at high temperatures. Noji et al. [78] call this transition the “strictly supersaturation-dependent transition” or “S transition”. Proteins exhibiting S transitions include those with a native conformation (β2m, variable (V_L_) and constant (C_L_) domains of antibody light chain, HEWL, and ribonuclease A (RNaseA)) and αSyn. Most interestingly, wild-type RNaseA forms amyloid fibrils upon heating under stirring. The hinge-loop-expanded mutant of RNaseA was reported to generate amyloid-like fibrils via 3D domain swapping, whereas the wild-type RNaseA did not [79,80]. Transthyretin (TTR) at pH 2.0 and 0.1 M NaCl can also be included in Type S, although TTR at a neutral pH tends to form amorphous aggregates [81]. 

Type A proteins: The second type of protein exhibits spontaneous amyloid formation at high temperatures even without agitation. Noji et al. [78] refer to this type of transition as “autonomous amyloid-forming transition” or “A transition”. Type A proteins include insulin, glucagon, islet amyloid polypeptide (IAPP), and Aβ(1-40). In other words, the high amyloidogenicity of these relatively short amyloid peptides does not exhibit intrinsic barriers preventing amyloid formation. 

Type B proteins: The third type of protein often produces amorphous aggregates at high temperatures without a lag phase. Noji et al. [78] call this transition the “boiled egg-like transition” or “B transition”. Type B proteins (i.e., TDP-43, tau, and ovalbumin (OVA) [82]) are relatively large, and it is possible that their overall amorphous characteristics include amyloid cores (β-spines), producing a “fuzzy coat” morphology [83].

Noji et al. [78] showed that these three types (S, A, and B) of transitions with distinct responses to heating can be located on a general aggregation phase diagram, based on the driving forces of precipitation and protein solubility [12,84,85] (Figure 4A). The S, A, and B transitions are indicated by green, orange, and purple arrows, respectively. This type of diagram is often used to illustrate the crystallization and amorphous precipitation of native proteins and, moreover, for solutes in general (Figure 1) [10]. 

Thus, the S, A, and B transitions represent those from below solubility (Region I) to the metastable region (Region II), the labile region (Region III), and the amorphous region (Region IV), respectively. These transitions indicate that, with an increase in the driving force of precipitation at high temperatures, the aggregation behavior followed exactly as expected for solutes in general [10]. 

In terms of the phase diagram of conformational states (Figure 4A), stirring or ultrasonication is a kinetic factor modifying the apparent phase diagram. It is likely that the boundary between the metastable and labile regions is shifted downward upon agitation, decreasing the barrier of supersaturation and inducing spontaneous amyloid formation [3]. 

To address the mechanism underlying the distinct amyloidogenic transitions, Noji et al. [78] examined the relationship between transition types (i.e., S, A and B types) and various factors which might determine these types (Figure 4B,C). It is evident that the total residue number (abscissa) is the most dominant factor in determining the different types. Then, the hydrophobic score showed a notable correlation with the distinct amyloid types. When viewed from the perspective of size and hydrophobicity (Figure 4B), the S proteins had a moderate size and moderate hydrophobicity, the A proteins had a short length and high hydrophobicity, and the B proteins had a long length and low hydrophobicity. 

Another important factor is the disulfide bond [84,86,87]. The reduction in disulfide bonds often reduces amyloidogenicity, as demonstrated for β2m: under acidic conditions, the S-type transition changed to the B-type. These functions of disulfide bonds suggest that a more appropriate scale for evaluating the different types of amyloidogenic proteins is based on the “conformational flexibility of the denatured state”. 

Although the effects of disulfide bonds in reducing conformational entropy have been addressed [73,88], they are in fact minor in comparison with intrinsic ∆*S*_conf_ (Figure 4C). More importantly, the disulfide bonds stabilize hydrophobic cores that persist in the denatured state and thus increase amyloidogenicity, as demonstrated for acid-denatured β2m [84,87]. Taken together, the synergetic effects of disulfide bonds (i.e., decreasing the intrinsic conformational entropy and stabilizing the hydrophobic cores) lead to a significant decrease in the flexibility of the denatured states. 

## 3. HANABI, an Ultrasonication-Forced Amyloid Fibril Inducer

### 3.1. Ultrasonication-Dependent Breakdown of Supersaturation

Ultrasonication, conventionally used for amplifying seed amyloid fibrils [62,89,90], is an effective agitation method that triggers the nucleation process [91,92,93,94,95]. By combining an ultrasonicator and a microplate reader, Umemoto et al. [96] developed the HANdai Amyloid Burst Inducer (HANABI) system, which enables high-throughput analysis of amyloid fibril formation [95]. With the HANABI system, ultrasonic irradiation was performed in a water bath; the plate was then moved to the microplate reader, and ThT fluorescence was monitored. These three processes were repeated automatically based on programmed time schedules. Moreover, the plate was moved along the x–y axes in sequence, to ultrasonicate the 96 wells evenly. 

Kakuda et al. [97] used the HANABI system to amplify and detect αSN aggregates with seeding activity from CSF, and investigated the correlation between seeding activity and clinical indicators. The seeding activity of CSF correlated with the levels of αSN oligomers measured by an enzyme-linked immunosorbent assay. Moreover, the seeding activity of CSF from patients with Parkinson’s disease was higher than that of the control patients. Notably, the lag time of patients with Parkinson’s disease was significantly correlated with the ^123^I-meta-iodobenzylguanidine (MIBG) heart-to-mediastinum (H/M) ratio, one of the most specific radiological features of Parkinson’s diseases and dementia with Lewy bodies. These findings showed that the HANABI assay can evaluate the seeding activity of CSF by amplifying misfolded α-synuclein aggregates.

Although the original HANABI system promoted ultrasonication efficacy, several challenges remained. In general, the acoustic field in a sample solution could not remain the same because of changes in temperature, the volume of the water, and the distribution of the dissolved gases in the water bath. Specifically, in the original HANABI system, the fluorescence signal was acquired from the upper surface of the microplate, which was significantly affected by water droplets on the microplate because of the high-power ultrasonication.

Nakajima et al. [98] developed a HANABI-2000 system with an optimized sonoreactor for the amyloid-fibril assay, which improved the reproducibility and controllability of the amyloid fibril formation [95] (Figure 5). First, the water bath was eliminated in order to achieve a reproducible analysis. A single rod-shaped ultrasonic transducer was placed on each sample solution in an assay plate. The resonant frequency of the transducer was 30 kHz, which was optimized for accelerating amyloid fibril formation [99]. Second, a microphone was placed below the assay plate to measure the acoustic intensity of each sample solution. The acoustic intensity measurement allows for the acoustic field in each well to be controlled by individually regulating the voltage and frequency of the driving signal applied to each transducer. Third, a photodetector was placed beneath the microplate to measure the fluorescent signal, which improves the signal-to-noise ratio of the fluorescence measurement because of the absence of the water bath.

Using the acid-denatured β2m monomer solution, Nakajima et al. [98] demonstrated that achieving identical acoustic conditions by controlling the oscillation amplitude and frequency of each transducer results in synchronized amyloid fibril formation behavior across 36 solutions with a coefficient of variation (CV) of 22% for a half-time (*t*_half_) (Figure 5C). 

They then succeeded in detecting 100-fM seeds at an accelerated rate. Moreover, they revealed that acceleration of the amyloid fibril formation reaction with the seeds is achieved by enhancing the primary nucleation and fibril fragmentation. These results suggested the efficacy of HANABI-2000 in the diagnosis of amyloidosis owing to the accelerative seed detection, and the possibility of further early-stage diagnosis even without seeds through the accelerated primary nucleation (i.e., the identification of susceptibility risk biomarkers [49]).

Nakajima et al. [99] studied the molecular mechanism underlying the ultrasonication-dependent acceleration of amyloid fibril formation. They showed that ultrasonic cavitation bubbles behave as catalysts for nucleation: the nucleation reaction is highly dependent on the frequency and pressure of acoustic waves and, under optimal acoustic conditions, the reaction-rate constant for nucleation increased by three orders of magnitude. A theoretical model was proposed to explain the markedly frequency- and pressure-dependent nucleation; in this model, monomers are captured on the bubble’s surface during its growth and are highly condensed by the subsequent collapse of the bubble, so that they are transiently exposed to high temperatures [99]. Thus, the dual effects of local condensation and local heating contribute to markedly enhancing the nucleation reaction. 

### 3.2. Comparison of Ultrasonication and Shaking on Breaking Supersaturation

Although both ultrasonication and shaking are effectively used to induce amyloid fibril formation and propagation, the difference between them remained unclear until recently [89,100,101,102,103]. Nakajima et al. [104] compared ultrasonication and shaking with respect to the morphology and structure of resultant β2m aggregates, the kinetics of amyloid fibril formation, and seed-detection sensitivity. They focused on *t*_half_, the time required for exhibiting half of the maximal ThT fluorescence, and constructed a heat map, which describes the phase diagram of β2m aggregation. The experimental results show that ultrasonication markedly promotes amyloid formation, especially in dilute monomer solutions; it also induces short-dispersed fibrils, and is capable of detecting ultra-trace-concentration seeds with a detection limit of 10 fM. 

Most importantly, they indicated that ultrasonication highly alters the energy landscape of an aggregation reaction due to the effects of ultrasonic cavitation. Under shaking (Figure 6B), the metastable region becomes narrower than that under quiescence (Figure 6A), showing that shaking induces a downward shift in the metastable−labile boundary, whereas it has only minimal effects on the labile−amorphous boundary. On the other hand, ultrasonication not only causes a significant downward shift in the metastable−labile boundary but also an upward shift in the labile−amorphous boundary (Figure 6C). 

In the labile region, although the acceleration ability of shaking was similar to that of ultrasonication for solutions with high monomer concentrations, it decreased for solutions with a monomer concentration lower than 0.1 mg/mL. The aggregation acceleration by shaking results from the increase in the apparent mean-free path of monomer movements. Thus, shaking enhances the probability of intermolecular interactions in a condensed solution by increasing the collision frequency among monomers. However, it fails to increase the collision frequency in a dilute solution, diminishing the acceleration effect for nucleation (Figure 6B). In contrast, ultrasonication retains a high acceleration ability, even for dilute monomer solutions, because the cavitation bubble works as a catalyst for the nucleation reaction [99]. On the other hand, if the bubble’s surface becomes fully covered with monomers, the acceleration effect saturates. This mechanism consistently explains why the *t*_half_ value of the ultrasonication-dependent acceleration cannot be decreased below the lower limit of ~10 h for solutions with high monomer concentrations. 

## 4. Liquid–Liquid Phase Separation

One of the most important phenomena related to amyloid formation is liquid–liquid phase separation, which is observed at increasing rates in disordered proteins [17,105,106]. There are cases in which amyloid formation is preceded by the liquid–liquid phase separation. As an example, the low-complexity domain of FUS protein formed phase-separated droplets before the formation of more stable amyloid fibrils [107]. These results are consistent with Ostwald’s ripening rule of crystallization, according to which the morphologies of crystals change over time, guided by their kinetic accessibilities and thermodynamic stabilities [108]. 

Recently, Shimobayashi et al. [17] examined the liquid–liquid phase separation of FUS protein in living cells with a light-controlled oligomerizing system Corelets [109] and constructed the core protein concentration-dependent phase diagram. Their results show that the initial stages of nucleated phase separation can be modelled by classical nucleation theory, which describes how the rate of droplet nucleation depends on the degree of supersaturation. It is noted that, in their phase diagram of polymer science, the “binodal boundary” corresponds to solubility or critical concentration and the “spinodal region” correspond to the amorphous region. We assume that the “macroscopic” phase diagram of conformational states, such as Figure 1, Figure 3, Figure 4 and Figure 6, will be also useful for understanding the liquid–liquid phase separation, where “microscopic” phase diagrams limited by supersaturation might apply to each droplet system. 

The relationship between phase-separated droplets and oligomers or relatively small amorphous aggregates is not clear. Amorphous aggregation and amyloid fibrillation have often been considered as separate pathways in direct competition with each other, such that accumulation of amorphous aggregates will always retard fibrillation [3,11,110,111,112]. On the other hand, Nitani et al. [31] suggest the presence of two types of aggregates, i.e., a frozen amorphous aggregation state, and a labile amorphous aggregate capable of slow fibrillation. The labile amorphous aggregate is consistent with Ostwald’s ripening rule of crystallization, in which morphologies of crystals change over time, guided by their kinetic accessibilities and thermodynamic stabilities [6,108]. Moreover, it is possible that spontaneous amyloid formation is assisted by a small amount of amorphous aggregates/amorphous oligomers, as suggested by the new view of crystallization [29].

## 5. Conclusions

Recent structural studies on amyloid fibrils have advanced remarkably, clarifying their atomic structures and polymorphisms [38,39,40,41,42]. However, amyloid structures do not necessarily fully explain the mechanism of their formation. Physicochemical studies on protein folding and misfolding have established that amyloid fibrils are crystal-like aggregates of denatured proteins, which are formed above solubility by breaking supersaturation [6,33,34,50]. It is evident that no amyloid formation occurs below solubility; moreover, the preformed amyloid fibrils dissolve below solubility [113,114], although the rigidity of amyloid fibrils may prevent rapid dissolution. 

Although the validity of Anfinsen’s dogma (i.e., reversible unfolding/refolding) is often questioned under high protein concentrations where intermolecular interactions are favored [115], the persistence of supersaturation and the difficulty of amyloid formation have neglected to address the question exactly. Recent studies that focus on solubility and supersaturation have shown that, although protein folding is independent of protein concentration, amyloid formation is critically dependent on protein concentration (i.e., solubility). By combining the concentration-independent folding/unfolding under supersaturation and the concentration-dependent amyloid formation after the breakdown of supersaturation, the exact unification of these two processes has become possible. In other words, the breakdown of supersaturation links Anfinsen’s intramolecular folding universe and the intermolecular misfolding universe, extending our understanding of protein folding and misfolding.

The conformational phase diagrams used in this article are similar to a general phase diagram of a solute consisting of soluble (Region I), metastable (Region II), labile (Region III), and amorphous regions (Region IV) (Figure 1, Figure 3, Figure 4 and Figure 6). The phase diagrams based on solubility and supersaturation will be essential for further clarification of the mechanisms of protein folding and misfolding, and their roles in diseases and life functions. 

Finally, in relation to supersaturation-dependent amyloid formation, the onset of amyloid deposition is accompanied by a decrease in the soluble concentration of precursor proteins, as discussed in this article [49,50]. In other words, monitoring decreases in precursor protein concentrations might be a promising approach for detecting the onset of amyloidosis.

## Figures and Tables

**Figure 1 molecules-27-04588-f001:**
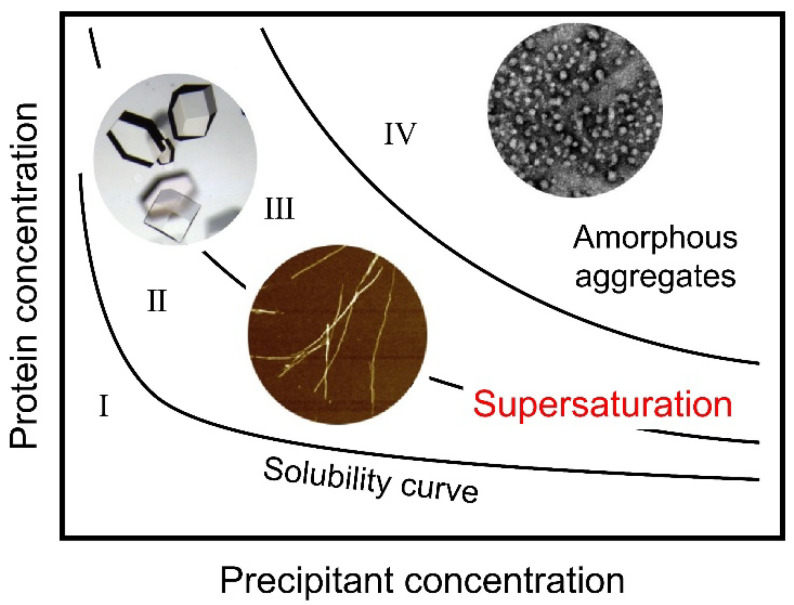
Protein and precipitant concentration-dependent phase diagram common to native crystals and aggregates of denatured proteins. Regions I, II, III, and IV represent undersaturation, the metastable region, the labile region, and the amorphous region, respectively. Crystallization and amyloid fibril formation occur from regions II and III. The figure was modified from So et al. [6], with permission. Copyright 2016 Elsevier.

**Figure 2 molecules-27-04588-f002:**
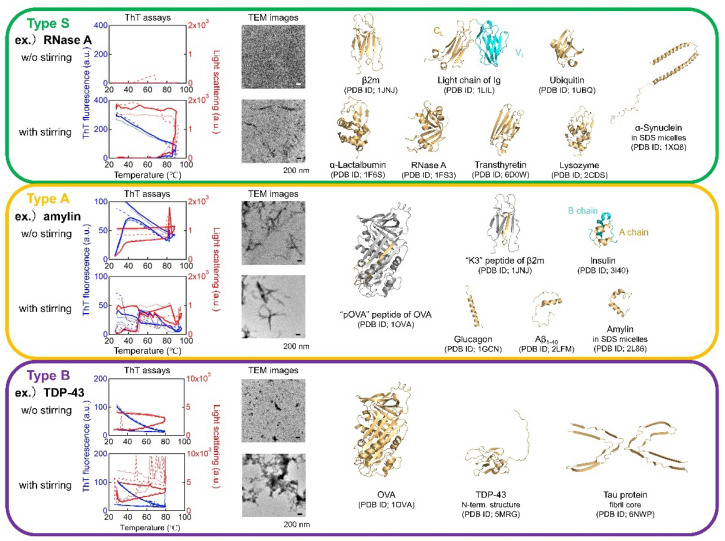
Heating- and agitation-dependent amyloid formation of proteins. According to their aggregation behavior, Type S, A, and B proteins were defined. **Left**: ThT assays upon heating in the presence (upper) or absence (lower) of stirring. The intensities of ThT fluorescence and LS are indicated by blue and red lines, respectively. *n* = 3. Middle: TEM images of the samples after heating in the presence (upper) or absence (lower) of stirring. **Right**: Structures of proteins used with their names and pdb codes. The figure was created based on Noji et al. [2].

**Figure 3 molecules-27-04588-f003:**
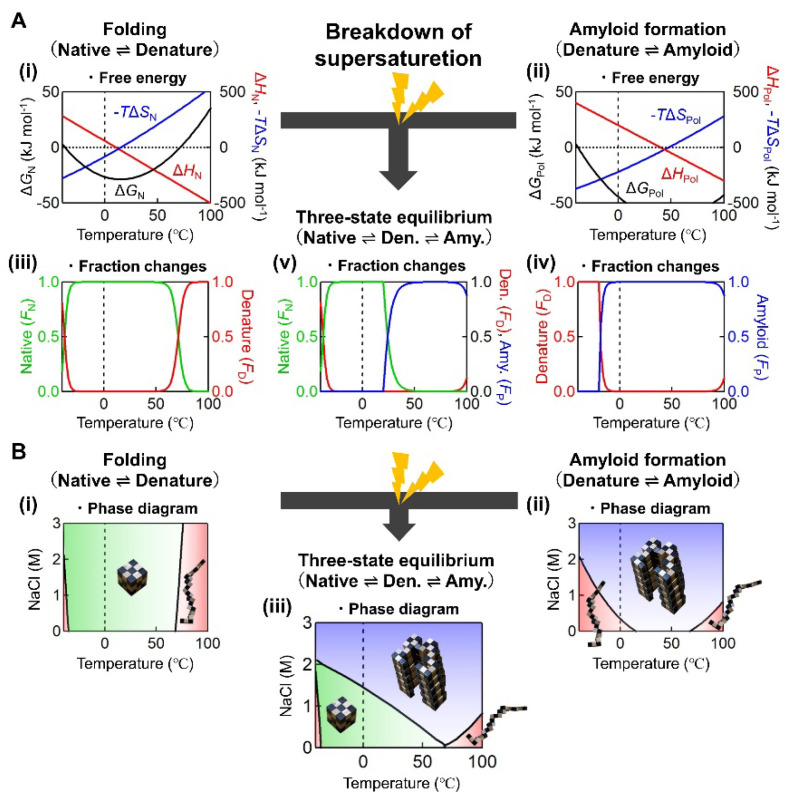
Temperature- and NaCl concentration-dependent conformational phase diagrams of β2m before and after the linkage of folding and misfolding transitions. (**A**) Temperature dependencies of thermodynamic parameters (Δ*G*(*T*), Δ*H*(*T*), and *T*Δ*S*(*T*)) for folding (Mechanism 1, panel (i)) and amyloid formation (Mechanism 2, panel (ii)). Fractions of N, D, and P states for folding (panel (iii)), amyloid formation (panel (iv)), and their linked conditions (panel (v)) are also shown. The plots were made using 0.1 mg/mL β2m, 1.0 M NaCl, and pH 7.0. (**B**) Phase diagrams for folding/unfolding (Mechanism 1, panel (i)), amyloid misfolding (Mechanism 2, panel (ii)), and their linked conditions (panel (iii)). Lines show the simulated phase boundaries at 0.1 mg/mL β2m and pH 7.0. The figure was modified from Noji et al. [12].

**Figure 4 molecules-27-04588-f004:**
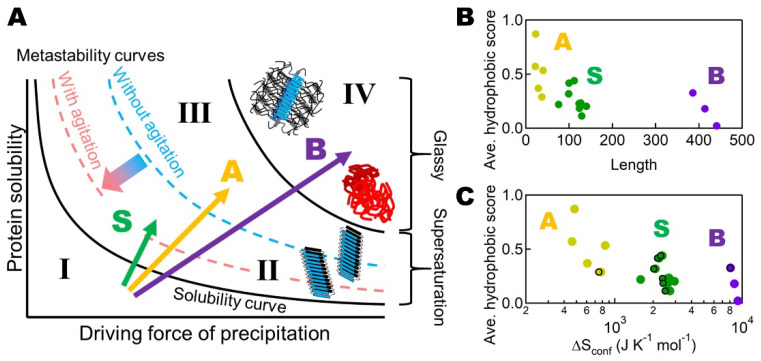
General schematic conformational phase diagram and the three transitions. Three types of amyloidogenic proteins were plotted on a general phase diagram of aggregation (**A**), and diagrams of average hydrophobicity vs. number of amino acid residues (**B**) or ∆S_conf_ (**C**). In c, ∆S_conf_ represents an increase upon denaturation of the main chain with (points within the frame) and without (points outside the frame) the contribution of disulfide bonds. The figure was reproduced from Noji et al. [78].

**Figure 5 molecules-27-04588-f005:**
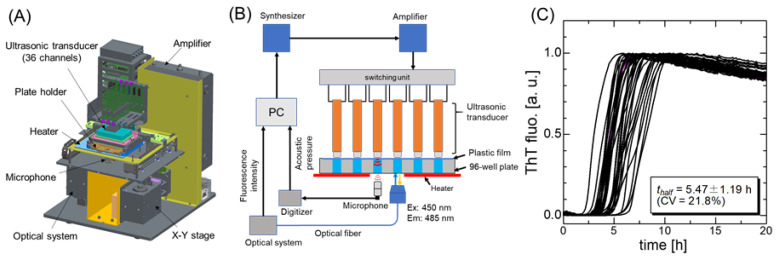
Overview of the HANABI-2000 system. (**A**) A 3D schematic illustration of the optimized sonoreactor for the amyloid-fibril assays, HANABI-2000. The dimensions of the device are 500 × 550 × 550 mm^3^. (**B**) A block chart of the control units of HANABI-2000. The figure is reproduced from Nakajima et al. [98] with permission. (**C**) The ThT time–course curves (*n* = 36), which are irradiated with ultrasound with the compensation procedure. The figure was modified from Nakajima et al. [98] with permission. Copyright 2021 Elsevier.

**Figure 6 molecules-27-04588-f006:**
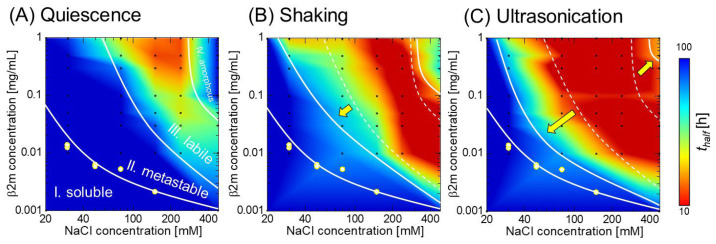
*t*_half_ heat maps of aggregation reactions. (**A**) Under quiescence, (**B**) under shaking, and (**C**) under ultrasonication. The yellow dots denote the solubility of acidic β2m monomer at each salt concentration, determined by ultracentrifugation and the ELISA assay. The dotted lines in panels (**B**,**C**) indicate the phase boundaries under quiescence, which are varied under agitation as indicated by yellow arrows. The figure is reproduced from Nakajima et al. [104] with permission. Copyright 2021 American Chemical Society.

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
