# Peer review of "Supersaturation-Dependent Formation of Amyloid Fibrils†"

_molecules, 2022, doi:10.3390/molecules27144588_

Round 1

Reviewer 1 Report

This is an interesting and timely review by one of the leading groups in the field of amyloid fibrils formation. The manuscript focuses on supersaturation as a critical kinetic condition in the formation of amyloid fibrils and onset of amyloidosis. For the purpose of this review, the authors main focus is on very recent articles that propose that breakdown of the supersaturation barrier as necessary to shift proteins to the amyloid pathway, linking protein folding and misfolding.

 Authors review also the HANABI system that enables high-throughput analysis of amyloid fibril formation by the ultrasonication-triggered breakdown of supersaturation, and the optimized HANABI-2000 underlining the system high potentiality in early-stage diagnosis.

 Minor revisions:

-Conclusions can be improved: lines 359-365 can be relocated in the manuscript. Line 369: “Our results” should be removed and the paragraph differently introduced.

-Line 102: replace “namely that the proteins..” with “stating that the proteins..”

Author Response

See the file attached.

Reviewer 2 Report

Goto et al. present a review paper on the role of supersaturation during amyloid fibrillation that updates a 2016 review on the same subject (ref. [3]). Although the paper is very much focused on the results obtained by the authors’ groups, I found it very interesting and well written. I have some concerns about the accuracy of some important concepts and, for this reason, I cannot recommend the present version of the manuscript for publication. I look forward to changing my opinion after the following remarks are addressed:

1. Lines 12 and 33: The definition of supersaturation presented by the authors is not accurate. Supersaturation depends on protein concentration, solubility, temperature and pH, but is independent of kinetic parameters such as nucleation and growth rate constants. As such, supersaturation does not refer to a kinetic condition. Rather, it measures how far from the thermodynamic equilibrium a soluble protein is in a given environment. “An apparently soluble state” is also a highly misleading expression. The state of a protein is either in the soluble or insoluble phase, independently of whether the soluble fraction is supersaturated or not. Kinetic trapping in a soluble state is generally used to define metastability (which is different from the supersaturated state). Likewise, only metastability is “broken” once crystallization starts, whereas the supersaturated state remains long after the onset of phase separation. For the same reason, “supersaturation breakdown” is a misleading expression. Please correct these issues.

2. Lines 44 to 46: Following the previous remark, I think that the physicochemical principles underlying the supersaturated state are well established and not “elusive”. The authors should specify what aspects of supersaturation are elusive. I could find none in the cited literature [1, 7-9]. Is it possible that the authors are thinking about the mechanisms of nucleation when using the adjective “elusive”?

3. Lines 47/48. This sentence needs rephrasing: The phase diagram is not something presented “here” since many similar phase diagrams have been presented before. Also, only the location of the supersaturated region is illustrated by the phase diagram – not its “role”.

4. Lines 64 to 70 are textually taken from ref. [3]; the whole first paragraph of section 2 is identical to the first paragraph of Section “Supersaturation in amyloid fibrillation” in ref. [3]. I do not find this copying-and-pasting appropriate.

5. Lines 142 to 173. In a review paper about supersaturation-dependent amyloid fibrillation, it is odd that the only biophysical mechanism that is presented relies on chemical equilibrium principles and not on phase equilibrium ones. It is also strange that a definition of supersaturation is also missing either in terms of the variation of chemical potential or, more simply, in terms of protein concentration and solubility. Please correct and add a physicochemical definition of supersaturation.

6. Conclusions, first paragraph: The first paragraph of Conclusions discusses liquid-liquid phase separation in the context of supersaturation-dependent phase separation. This is an interesting topic but was not addressed anywhere else in the manuscript. I suggest including this paragraph in a small subsection of Discussion entitled Future Challenges or similar. It would be interesting to include a word about non-amyloid oligomers, fibril polymorphism, biological polymers and biological condensates, for example.

- Conclusions, second paragraph: In a review paper, it is not common to limit the scope of the Conclusions only to results obtained by the authors. I would remove expressions such as “our results show…” or “our consideration indicates”.

Author Response

See the file attached

Reviewer 3 Report

Goto et al. prepared a manuscript reviewing the mechanism and role of supersaturation in protein misfolding and fibrilization. Through reading this review article, the readers will gain a throughout understanding of the mechanism of supersaturation-dependent amyloid formation and ultrasonication-based fibril induction. This manuscript allows readers to pay more attention to the role of supersaturation in amyloidosis. However, there are some limitations preventing this manuscript from being a comprehensive review article:

(1) The abstract states that supersaturation-dependent fibril formation is important for therapeutic development and structural studies, but these were not addressed in the main text. I would recommend discussing how supersaturation-dependent fibril formation is crucial in therapeutic development and structural studies in the review, to make it more appealing to general readers. 

(2) The authors reviewed many good works from their previous publications, but this manuscript is too narrowed to the work from the same group of people. To make this review more comprehensive, the authors may consider adding more literature evidence from other research groups about the role of supersaturation in protein unfolding/folding and fibril formation. It may be nice to comment on other leading researchers’ work in the area of supersaturation. 

Some minor comments:

(3) “Supersaturation” should be one of the keywords.

(4) If “β2-microglobulin” is one of the keywords, why not “amyloid-β”?

(5) Figure 5C: explain in the figure caption what does the color code mean (blue lines vs purple lines)

(6) Line 49: “region I” should be used instead of “region 1”.

(7) Line 187-213: line spacing is inconsistent with others.

Author Response

See the file uploaded.

Round 2

Reviewer 2 Report

I would like to thank the authors for the introduced modifications and for improving their original version. A misunderstanding of the concept of “supersaturation” seems, however, to persist. I reiterate that it is erroneous to consider supersaturation as a “phase” in which “the solution is trapped in a soluble state”. The authors could not identify an independent bibliographic source supporting their definition simply because this definition is wrong. The definitions taken from the book by J.W. Mullin and the review paper by Coquerel are perfectly fine. Supersaturation is indeed the driving force for crystallization, which is utterly different from saying that supersaturation is a “phase”, or “trapped soluble state” or a “phenomenon”. The authors mention a “new” view of supersaturation as a kinetically trapped state using as reference a 1962 paper by Oosawa and Kasai. I could not find a single reference to supersaturation or kinetically trapped states in the said paper.

I consider, therefore, that the present version of the manuscript is stained with a highly misleading definition of supersaturation and should not be published.

Author Response

Our responses to Reviewer 2 (Second round):

The reviewer reiterates that it is erroneous to consider supersaturation as a “phase” in which “the solution is trapped in a soluble state”. However, the reviewer does not describe his/her definition or underlying mechanism of supersaturation. Considering the comment, we revised the introduction.

We already defined supersaturation in the previous version (lines 34-37): “supersaturation of a solution refers to a non-equilibrium phase in which, although the solute concentration is higher than the thermodynamic solubility, the solute molecules remain soluble for an extended period because of a high free energy barrier to nucleation”.

We contrasted the classical and new views of supersaturation (lines 72-76): The physicochemical mechanisms underlying supersaturation have been studied extensively. “One of classical mechanisms underlying supersaturation is the difficulty of nucleation as modeled for actin polymerization by Oosawa and Kasai [16] or classical nucleation theory [17] (lines 72-75). However, subsequent studies suggest a more complicated mechanism of supersaturation, in which solutes form a kinetically trapped state, which is located on the distinct pathway(s) to formation of crystals”.

It is noted that although Oosawa and Kasai [16] never used supersaturation, as the reviewer commented, their model has been considered to model supersaturation-dependent actin polymerization as cited by the review article of Morris, et al. [14].

It is also noted that a new view of crystal nucleation has been reported, which is already described in the previous version (lines 89-100): “a recent view of crystal nucleation focuses on highly concentrated disordered droplets observed before crystallization [4, 5, 27, 28]”.

Then, we concluded the introduction with a sentence (lines 100-102) “Taken together, although supersaturation is a clear phenomenon, underlying physicochemical mechanisms how supersaturation develops, retains and is broken and, moreover, roles in crystal nucleation are still far from clear.” We hope these revisions clarify the concerns of Reviewer 2.

Reviewer 3 Report

As pointed out by other reviewers and myself, the original version of the review focused too much on the authors’ own work. In the revised version, the authors have addressed our comments and included many related works from other research groups. The review now is comprehensive with sufficient literature references.

The conclusion section is also improved and now highlights the importance of supersaturation-dependent fibril formation in therapeutic development and structural studies.

Figure 5C is also improved since the meaningless distinct color lines are no longer used.

I therefore recommend the current version of the manuscript be published in Molecules

Author Response

Response to Reviewer 3 (Second Round)

Thank you for your positive evaluation.